# Carbon sequestration: counterintuitive feedback of plant growth

Juan Alonso-Serra 

Laboratoire de Reproduction et Développement des Plantes, ENS de Lyon, Institut National de Recherche pour l'Agriculture, l'Alimentation et l'Environnement (INRAE), CNRS, 46 Allée d'Italie, 69364 Lyon Cedex 07, France.

## Insights

Plants; Soils; Carbon feedback.

**Author for correspondence:**
J. Alonso-Serra,
E-mail: juan.alonso-serra@ens-lyon.fr

## Abstract

Interaction between the atmosphere, plants and soils plays an important role in the carbon cycle. Soils contain vast amounts of carbon, but their capacity to keep it belowground depends on the long-term ecosystem dynamics. Plant growth has the potential of adding or releasing carbon from soil stocks. Since plant growth is also stimulated by higher $CO_2$ levels, understanding its impact on soils becomes crucial for estimating carbon sequestration at the ecosystem level. A recent meta-analysis explored the effect $CO_2$ levels have in plant versus soil carbon sequestration. The integration of 108 experiments performed across different environments revealed that the magnitude of plant growth and the nutrient acquisition strategy result in counterintuitive feedback for soil carbon sequestration.

Photo by Jim Richardson. Collection: Feeding the planet – Soils.

Understanding carbon exchange between the atmosphere and ecosystems provides valuable information for long-term decision-making. An important piece of this complex puzzle is estimating how terrestrial and oceanic environments will respond to higher $CO_2$ levels. Despite discrepancies in magnitude, a generally accepted model predicts that plant growth will increase in response to elevated atmospheric $CO_2$ levels (Walker et al., 2021). However, plants also modify the environment they grow in through multiple interactions that are difficult to generalise. For example, ecosystem-level carbon sequestration depends on how plant growth and species composition affect soils, which can result in very different scenarios. A recent study analysed more than a hundred ecosystem-level experiments with enriched $CO_2$ (e$CO_2$) to uncouple the main components of this variation (Terrer et al., 2021).

The path of carbon from the atmosphere to plants and then from plants to soils includes a series of ramifications, the magnitude of which is currently being studied and debated. According to the latest carbon budget, over the last decade ~31% of $CO_2$ emissions were fixed by terrestrial ecosystems, ~23% by the oceans, and the rest remained in the atmosphere (Friedlingstein et al., 2020). On land, nearly half of the carbon fixed each year is allocated to belowground systems such as roots, whereas the second half is allocated to aboveground growth (Gherardi & Sala, 2020). The process of carbon allocation to soils happens predominantly through root growth and rhizodeposition, though a portion of the carbon is also transferred to heterotrophic microorganisms that symbiotically benefit from valuable plant sugars. The combined effects of time, plant litter and plant death eventually lead to most terrestrial carbon stocks being sequestered not in plants but instead in soils. Globally, vegetation is estimated to contain 450–650 gigatons of carbon (GtC), while 1,500–2,400 GtC are sequestered in soils and ~1,700 GtC in the permafrost (Friedlingstein et al., 2020). Therefore, while plant growth plays an essential role in carbon fixation, the importance of plants is primarily their function as necessary tunnels and dams of long-term underground storage.

If this were a one-way pathway, increased plant growth stimulated by $eCO_2$ would always result in larger soil carbon stocks. However, a restrictive feedback known as the *priming effect* counteracts this: plant growth can also stimulate the decomposition of soil organic matter, thereby releasing soil $CO_2$ to an extent that may compromise net ecosystem-level carbon fixation (Kuzyakov et al., 2000; van Groenigen et al., 2014).

The meta-analysis performed by Terrer et al. revealed that while $eCO_2$ generally has a positive effect on aboveground plant biomass, its effect on soil organic carbon (SOC) is variable. The authors showed that there is an inverse relationship between carbon stocking in plants versus soil in unfertilised soils; the stronger $eCO_2$ stimulates plant biomass growth, the more SOC stocks tend to diminish. By contrast, mild $eCO_2$ levels promote both plant biomass and SOC accumulation.

Many local variables can affect this dynamic, but a critical factor in quantitative terms seems to be nutrient availability. When aboveground plant growth is strongly stimulated by $eCO_2$ in nutrient-scarce conditions, the roots need to mine the soil further, thereby accelerating the decomposition of organic matter. In this case, enhanced plant growth stimulates increased release of soil $CO_2$ originating from microbial respiration, resulting in a considerable reduction in SOC stocks. When nutrients are accessible, this effect can be dampened but not necessarily reverted, as was seen in nitrogen-fertilised soils.

This seems to imply that soil nutrients are the determinant factor for plant-based carbon sequestration, but the richness of ecosystem interactions again suggests a more complicated picture. Soil composition varies not only in terms of nutrients, but also in the communities of plants, animals and microorganisms living in the soil. Dramatic differences in plant nutrition can result from the activity of plant-symbiotic fungi known as mycorrhiza. The two largest groups are ectomycorrhiza (ECM), more frequently found in boreal and temperate forests, and arbuscular mycorrhiza (AM), predominantly present in tropical forests and grasslands (Soudzilovskaia et al., 2019). Mycorrhiza facilitate root access to valuable nutrients for plant growth, such as nitrogen and phosphorus. In exchange, plants provide both types of mycorrhiza with significant amounts of photosynthesised sugars.

Terrer et al. found that in ecosystems with significant ECM associations, an $eCO_2$ environment promotes nitrogen uptake and plant growth both above and below ground, which reduces SOC stocks due to the priming effect. However, AM associations do not result in a strong link between $eCO_2$ and nitrogen uptake, and $eCO_2$ therefore has only a moderate effect on plant growth, resulting in the steady accumulation of SOC through fine-root growth and rhizodeposition. This feedback is important, because in boreal ecosystems the priming effect can be so strong that it would result in virtually no carbon sequestration even after more than 30 years of plant growth (Friggens et al., 2020). A similar feedback was also observed in $eCO_2$ experiments in warm-temperate forests (Jiang et al., 2020).

These examples illustrate the delicate site-specific equilibrium that must be considered when plants are used in a carbon sequestration strategy. Plants can certainly sequester carbon, but reaching a carbon negative condition at the ecosystem level depends on additional variables. Water availability, temperature and plant pathogens are also essential to the ecosystem equation due to their direct effect on plant growth. In addition, other important greenhouse gases such as methane cycle between natural ecosystems and the atmosphere (Saunois et al., 2020). Therefore, the importance of ecosystems is not only their carbon sponge capacity, but also their ability to lock carbon underground, an ability that is critically compromised by land use disturbances.

Finally, it is worth highlighting that long-term decision-making should assess the contribution of plants and ecosystems not only in terms of carbon budget, but also in the multiple services they provide to species that interact with them, including us. These studies invite us to think of plant growth not only as a carbon sink, but also as a transitory stage of carbon and to keep in mind the counterintuitive complexity of ecosystem feedback.

## Acknowledgements

The author is grateful to Professor Jaana Bäck and Docent Jussi Heinonsalo for critically reading this manuscript. The author thank Seeder el-Showk for proofreading the text and Olivier Hamant, Satu-Emilia Myllymäki and Marianne Lang for stimulating this discussion. The author also thanks Jim Richardson for providing the photo.

**Financial support.** This work was supported by the EMBO Long Term Postdoctoral Fellowship (ALTF 1-2020).

**Conflicts of interest.** The author declares no conflict of interest.

**Authorship contributions.** This manuscript was conceived and written by Juan Alonso-Serra.

**Data availability statement.** No data or code were developed for this manuscript.

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
