## [Reviewer Report]

Dear Editor,

Please find attached my manuscript entitled “Counterintuitive feedbacks of plant growth”, which I wish you to consider as an “Insight” for Quantitative Plant Biology.

In this manuscript, my aim is to highlight a recent publication addressing the effects plant growth has on soil carbon stocks (1). With the surge of new technologies and climate mitigation strategies aiming to increase plant yields and carbon sequestration, I find it timely and relevant to highlight the counterintuitive nature of ecosystemic feedbacks. The recent article published by Terrer et al is a perfect example of such complexity. The authors studied the effect CO2 levels have in plant vs. soil carbon sequestration across more than a hundred different environments. They show that the stronger CO2 stimulates plant biomass growth, the more soil carbon stocks tend to diminish. By contrast, mild CO2 levels promote both plant biomass and soil carbon accumulation. 

I graduated from my doctoral studies in September 2020. Although my scientific expertise is in the field of plant development, as an early career researcher I think is important to broaden the scientific discussion and consider also the impact plants may have on their environment. This, I believe, will better inform us towards long-term solutions of carbon sequestration.

Sincerely,

Juan Alonso-Serra

Postdoctoral Researcher

RDP - INRAE - ENS de Lyon

46 allée d'Italie

69007 Lyon

France

(1) Terrer, C., Phillips, R. P., Hungate, B. A., Rosende, J., Pett-Ridge, J., Craig, M. E., van Groenigen, K. J., et al. (2021). A trade-off between plant and soil carbon storage under elevated CO2. Nature, 591(7851), 599–603.

---

## [Reviewer Report]

*Comments to Author*: This article was a simple review regarding the effect of elevated CO2 on soil carbon stocks on the results from many published papers. The review points should be OK in this article. However, elevated CO2 also affected CH4 and N2O emissions. Specially, CH4 emission is in the C budget, it should be noted in the last conclusions too.

---

## [Reviewer Report]

Dear Editor,

Please find attached the revised version of my Insight manuscript.

In response to the reviewer request, I have amended the text in lines 95-97. I also include a photo kindly provided by Jim Richardson for this publication.

In the text, I did the following minor modifications to improve the text flow and precision.

-Line 36: Removed the word “complex”

-Line 47: added “ ~ ” symbol

-Line 50: added “and dams of”

-Line 99: Removed sentence “Predicting global feedbacks and mechanisms is even more difficult because of uncertainties about ecosystem structure, capacity, size and behaviour”

-Line 140: Changed the word “humans” for “us”

-Acknowledgments: I included the acknowledgement to Jim Richardson for the photo.

Sincerely,

Juan Alonso-Serra